# Effects of Aphrodite (an Herbal Compound) on SSRI-Induced Sexual Dysfunctions and Depression in Females with Major Depressive Disorder: Findings from a Randomized Clinical Trial

**DOI:** 10.3390/medicina59091663

**Published:** 2023-09-14

**Authors:** Nasrin Shahmoradi, Omran Davarinejad, Annette Beatrix Brühl, Serge Brand

**Affiliations:** 1Department of Psychiatry, Kermanshah University of Medical Sciences (KUMS), Kermanshah 6714673159, Iran; odavarinejad@gmail.com (O.D.); annette.bruehl@upk.ch (A.B.B.); 2Center for Affective, Stress and Sleep Disorders, Psychiatric Clinics of the University of Basel, 4002 Basel, Switzerland; 3Substance Abuse Prevention Research Center, Kermanshah University of Medical Sciences (KUMS), Kermanshah 6714673159, Iran; 4School of Medicine, Tehran University of Medical Sciences (TUMS), Tehran 1417466191, Iran; 5Sleep Disorders Research Center, Kermanshah University of Medical Sciences (KUMS), Kermanshah 6714673159, Iran; 6Division of Sport Science and Psychosocial Health, Department of Sport, Exercise and Health, University of Basel, 4052 Basel, Switzerland; 7Center for Disaster Psychiatry and Disaster Psychology, Psychiatric Clinics of the University of Basel, 4002 Basel, Switzerland

**Keywords:** major depressive disorder, sexual dysfunctions, anxiety, sleep disturbances, herbal medicine, Aphrodite, ginger, saffron, thistle, cinnamon, *Tribulus terrestris*, sertraline, SSRI-induced sexual dysfunction

## Abstract

*Background and Objectives:* Almost by default, people with major depression disorder (MDD) also report sexual health issues. This holds even more true when sexual dysfunctions are SSRI-induced. Herbal compounds may have the power to counterbalance such sexual dysfunctions, though research is still scarce. Therefore, we assessed females with diagnosed MDD treated with a standard SSRI (sertraline) and reporting SSRI-induced sexual dysfunctions, and we asked whether compared to placebo, Aphrodite (a blend of ginger, saffron, cinnamon, thistle, and *Tribulus terrestris*) may favorably impact on sexual dysfunctions, and on symptoms of depression, anxiety, and sleep disturbances. *Materials and Methods:* A total of 41 females (mean age: 35.05 years) with diagnosed MDD, treated with sertraline (a standard SSRI) at therapeutic dosages, and reporting SSRI-induced sexual dysfunction, were randomly assigned either to Aphrodite or to the placebo condition. At baseline and four and eight weeks later (study end), participants completed a series of self-rating questionnaires covering symptoms of sexual dysfunction, depression, anxiety, and sleep complaints. *Results:* Symptoms of sexual dysfunction, depression, and anxiety decreased over time, but more so in the Aphrodite condition, compared to the placebo condition (significant *p*-values and large effect sizes). Over time, sleep disturbances decreased irrespective of the study condition. *Conclusions:* The pattern of results suggests that compared to placebo, Aphrodite appeared to improve symptoms of sexual dysfunction, depression, and anxiety among females with diagnosed MDD and SSRI-induced sexual dysfunction. Further and similar studies should investigate the underlying psychophysiological mechanisms.

## 1. Introduction

Among others, major depressive disorder (MDD) is associated with feelings of sadness, sleep disorders, suicidal behavior, slowed cognitive processing, decreased self-esteem, and an observable decrease in sexual interest, including sexual pleasure and sexual functioning [1,2,3].

Further, it was estimated that by 2020, MDD would have been the second most common disease after cardiovascular diseases [3]. In this line, a national study in Iran showed that the prevalence rates of MDD were almost 20%, and MDD was considered one of the most impacting health problems in Iran [4].

To treat MDD, although the primary approach involves the use of antidepressant medications, particularly selective serotonin reuptake inhibitors (SSRIs), recent reviews and meta-analyses have raised doubts about their effectiveness [5,6,7]. As a result of such a lack of effectiveness and of reported concomitant side effects, individuals with MDD may discontinue their medication treatment. As an alternative to antidepressants, other treatment options include (internet-delivered) cognitive–behavioral interventions [8,9,10,11,12], regular physical activity [13], electroconvulsive therapy [14,15,16,17,18,19], and the intake of omega-3 polyunsaturated fatty acids [20]. Additionally, older individuals, in particular, may be hesitant to use synthetic drugs and instead rely on herbal products such as John’s Wort [21].

In the present study, we focused on the possible impact of Aphrodite^®^, an herbal compound consisting of ginger, saffron, cinnamon, thistle, and tribulus terrestris, on symptoms of sexual dysfunction, depression, anxiety, and sleep disturbances, as compared to a placebo. To this end, we assessed females with MDD, undergoing treatment with sertraline (an SSRI) and reporting SSRI-induced sexual dysfunctions.

Among the antidepressants, selective serotonin reuptake inhibitors (SSRIs) appear to be highly prescribed [22,23]. Typically, a person takes an SSRI for at least two months in order to evaluate the treatment response [23]. Several studies indicated that SSRIs appeared to have positive effects on symptoms of depression [24,25]. In contrast, typical side effects of SSRI are feeling agitated, shaky, or anxious; indigestion, diarrhea, or obstipation, loss of appetite, weight loss and dizziness. Most problematic from the psychosocial point of view, SSRIs unfavorably impact on sexual functions [26]. More specifically, individuals may experience a reduction in their overall interest in sex or find it challenging to become sexually aroused. This may lead to difficulties in initiating or maintaining intimate relationships [27,28]. Additionally, SSRIs can affect sexual function and orgasm. Some individuals may experience delays in achieving orgasm, or they may find it more difficult to reach climax. For some, the intensity of orgasm may be diminished, leading to a less satisfying sexual experience [29]. Not surprisingly, Rosenberg et al. [30] found that approximately 41.7% of men and 15.4% of women had discontinued treatment due to sexual side effects from psychiatric medications. Further, given that SSRIs may cause sexual dysfunction in 40–65% of individuals with MDD [31,32,33,34,35,36,37,38], these side effects may exacerbate depression and create barriers to continued treatment [39]. Given this background, and given that among adults a satisfactory sexual life is associated with an overall higher quality of life [31,32,33], research in non-synthetic compounds to improve female sexual behavior is justified.

Medicinal herbs and herbal medicinal formulations are commonly prescribed for the treatment of mental problems [34,35,36,37]. In parallel, the number of individuals using herbs for their health is increasing, probably because of a skeptical attitude towards the effectiveness of synthetically produced compounds and their concomitant side effects [38]. Studies showed the mechanisms of action of plants used to treat depression, anxiety, and sleep disorders, which include the reuptake of monoamines, effects on neuroreceptor binding and neurotransmitter activity, and the modulation of the neuronal communication or of the hypothalamic–pituitary adrenal (HPA) axis [39,40]. However, research on the effectiveness of herbal compounds in women with sexual disorders caused by the use of antidepressants is scarce and inconclusive. While compared to placebo, *Rosa damascena* improved methadone-induced sexual dysfunctions in females with opioid use disorder [41,42], *Rosa damascena* had only a small effect on SSRI-induced sexual dysfunctions among females with diagnosed major depressive disorder [43]. Given these conflicting results, we hold that it was important and necessary to examine herbal medicine on the level of depression, anxiety, and sleep disorders [42].

Next, sleep quality plays a pivotal role in the intricate interplay between depression and sexual functioning [9,20,44]. Sleep disturbances are not only common symptoms of depression but also exert a profound impact on sexual health. The relationship between depression and sleep disruption is bidirectional. Individuals with depression often experience disrupted sleep patterns, while poor sleep quality can contribute to the development or exacerbation of depressive symptoms [11,25,26]. Similarly, sexual functioning may be significantly affected by both depression and sleep disturbances. Depression may lead to reduced sexual desire, arousal difficulties, and overall sexual dissatisfaction. Moreover, disrupted sleep patterns may directly impede sexual function, leading to fatigue, diminished libido, and challenges in achieving and maintaining arousal [3,9,13,27,28,29]. Given this, the decision was to further assess symptoms of sleep disturbances along with sexual dysfunctions and symptoms of depression.

As mentioned, one possibility to additionally improve psychological health is the application of complementary and alternative medicine [45]. One of the herbal supplements is Aphrodite, a combination of several herbs such as ginger, saffron, cinnamon, thistle, and *Tribulus terrestris* [46]. This herbal compound is known for its rich content of bioactive substances such as crocetin and crocetin glycosides, which contribute to numerous physiological benefits that support both mental and physical health [47]. Specifically, ginger, saffron, cinnamon, thistle, and *Tribulus terrestris* possess anti-inflammatory, antioxidant, and neuroprotective properties. While it is thus conceivable that the sum of these compounds has an important effect in the prevention and treatment of various diseases [48,49], it remained unclear whether the effect is isolated or the combined effect of the components. Additionally, the above-mentioned herbs appeared to improve cognitive function, reduce stress and anxiety, promote digestion, enhance immune response, and support cardiovascular health [50,51,52]. Given this, such a broad range of bioactive compounds makes Aphrodite a valuable add-on to a healthy lifestyle, offering a holistic approach to overall well-being. More specifically, there is evidence that Aphrodite favorably impacted on sexual functions [53,54,55]. To illustrate, Taavoni et al. [56] showed that Aphrodite was effective in orgasm and sexual desire in postmenopausal women. This study involved 80 postmenopausal women aged 50–60 years and utilized a randomized clinical trial design with a control group. The results showed that following one month of intervention, the Aphrodite group demonstrated a significant increase in the orgasm score compared to the placebo group. The findings suggested that Aphrodite appeared to have a beneficial impact on sexual behavior among postmenopausal women.

### The Current Study

The rationale of the present study was as follows: First, individuals with MDD in general, and females with MDD, more specifically, suffer from sexual dysfunctions. Second, to treat symptoms of depression SSRIs are very often prescribed, though, third, a major adverse side effect of SSRIs is to impair sexual function. Fourth, sexual functioning is considered an integral part of psychological well-being and satisfactory couple life [5,31,32,33,44,57]. Fifth, symptoms of depression, sexual dysfunction, and sleep difficulties are highly intertwined. Seventh, the general population considers herbal compounds as more secure and with fewer side effects. Eighth, research on herbal compounds on sexual function among females with MDD is scarce and conflicting [35,41,43]. Given this, the decision was to investigate the impact of Aphrodite on symptoms of sexual dysfunction, depression, anxiety, and sleep disturbances, as compared to placebo. To this end, we assessed females with MDD, treated with sertraline (SSRI) and reporting SSRI-induced sexual dysfunctions, and randomly assigned them either to the Aphrodite or to the placebo condition.

Following the research mentioned so far, the hypothesis was that compared to placebo, the administration of Aphrodite improved symptoms of sexual dysfunction, depression, anxiety, and sleep disturbances.

## 2. Method

### 2.1. Procedure

The current study was a randomized, double-blind, and placebo-controlled clinical trial among females with MDD and SSRI-induced sexual dysfunction, undergoing a standardized treatment with sertraline at therapeutic dosages, and assigned either to the Aphrodite or to the placebo condition. From May to July 2023, eligible females with MDD, taking sertraline for at least six weeks, and routinely treated as out-patients of the Farabi Hospital (Kermanshah, Iran) were approached to participate in the present study. After a thorough clinical interview, including the examination of inclusion and exclusion criteria (see below), participants were fully informed about the study aims and about the confidential and secure data handling. Thereafter, participants signed the written informed consent. At baseline and four and eight weeks later (study end), participants completed a series of self-rating questionnaires covering symptoms of sexual dysfunction, depression, anxiety, and sleep complaints (see below). The Review Board of Kermanshah University of Medical Sciences (Kermanshah, Iran) approved the study, which was conducted in accordance with the ethical principles in the Declaration of Helsinki and its later amendments [58] (registration of a clinical trial: IR.KUMS.REC.1400.230; 6 March 2022.)

### 2.2. Randomization and Sample Size Calculation

For the randomization of participants, a computer-generated random number sequence was prepared, and tickets were consecutively numbered, placed in a ballot box, and drawn by an independent researcher not further involved in the study to determine the condition of the participants.

A power analysis using G*Power 3.1 [58] software (Düsseldorf, Germany) indicated 32 participants were required (16 per group) to detect an effect of moderate magnitude (f = 0.25; α-error = 0.05, power = 0.95, groups = 2, number of measurements = 3, correlation among general linear measures: r = 0.50). To counterbalance possible dropouts, the total sample size was set at 54 individuals (=27/condition), and 41 patients completed the study (see Figure 1).

### 2.3. Participants

Inclusion criteria were as follows: (1) age between 18 and 50 years; (2) female gender; (3) married and self-reporting a stable long-term heterosexual relationship; (4) regular menstruation; (5) diagnosis of major depressive disorder, as ascertained by experienced psychiatrists and clinical psychologists via a thorough clinical interview [59] to diagnose psychiatric disorders following the DSM-5 [60]; (6) willing and able to comply with the study conditions; (7) undergoing a standardized treatment with sertraline at therapeutic dosages of 50 mg/d at least six weeks before entering to the study; (8) reporting sexual dysfunctions plausibly and thus causally related to the regimen of sertraline intake (see below); (9) scores of 26.55 or lower in the Female Sexual Function Index (FSFI) (see below); (10) signed written informed consent. Exclusion criteria were as follows: (1) further psychiatric disorders such as bipolar disorders, substance use disorders, personality disorders, eating disorders, symptoms of anxiety, PTSD, adjustment disorders or states of psychosis; (2) acute suicidality; (3) chronic physical diseases such as hypertension, diabetes and rheumatism; (4) use of any medication that could have affected the patient’s sexual activity such as liver and kidney disease; (5) any known sensitivity to ginger, saffron, cinnamon, thistle, and *Tribulus terrestris*; (6) history of any glandular disease such as high prolactin and thyroid disorders; (7) reported and observable change in sexual behavior due to unexpected experiences such as planning pregnancy or sexual harassment; (8) drugs and substances such as methadone, heroin, or morphine known for their impact on female sexual functions; (9) pregnant or breastfeeding; (10) self-declaring to be in peri-menopausal state; (11) withdrawal from the study.

### 2.4. Compounds

#### 2.4.1. Aphrodite

Aphrodite tablets were purchased from Gol Daru Company (Tehran, Iran). Each tablet of Aphrodite contained ginger (12.27 mg), saffron (3 mg), cinnamon (11 mg), thistle (14 mg), and tribulus terrestris (40 mg). Two Aphrodite tablets were taken once a day in the evening.

#### 2.4.2. Placebo

Placebos consisted of farina and were provided by the same producer as the Aphrodite tablets (Gol Daru Company; Tehran, Iran). Placebo and Aphrodite tablets were identical in shape, weight, color, and scent. Both participants and staff members involved in the study were unaware of the Aphrodite or placebo assignment. As for Aphrodite, two placebo tablets were taken once a day in the evening.

### 2.5. Measures

#### 2.5.1. Sociodemographic Information

Participants reported their age (years), anthropometric information (weight and height), civil status (married), highest educational degree (high school, bachelor and master’s), and current occupation (employment and non-employment).

#### 2.5.2. Assessing SSRI-Induced Sexual Dysfunction

To assess SSRI-induced sexual dysfunction, after the thorough psychiatric interview (based on a semi-structured interview [61] to assess psychiatric disorders following the DSM-5 [60]), psychiatrists also explored patients’ sexual dysfunction before starting the treatment with sertraline, and at least six weeks before patient entry to the study, as well as current sexual dysfunction. An SSRI-induced sexual dysfunction was diagnosed sensu DSM-5 [60], if, all else being equal, sexual dysfunction emerged with the start of SSRI intake.

#### 2.5.3. Sexual Functions

To assess sexual functions, participants completed the Farsi version [62] of the Female Sexual Function Index (FSFI) [63]. The questionnaire consists of six items focusing on sexual desire, arousal, lubrication, orgasm, sexual satisfaction, and pain (reverse-coded). Answers are given on six-point Likert scales ranging from 0 (=never) to 5 (=always), with higher sum scores reflecting higher sexual function. A cutoff total score of ≤26.55 has been proposed for diagnosis of female sexual dysfunction, such that any woman who scores less than 26.55 should be considered at risk for sexual dysfunction [64].

#### 2.5.4. Symptoms of Depression

To assess symptoms of depression, participants completed the Farsi version [65] of the Beck Depression Inventory (BDI) [66]. This questionnaire has 21 items addressing depressive symptoms, including apparent and reported sadness, inner tension, reduced sleep and appetite, concentration difficulties, lassitude, inability to feel, pessimistic thoughts, and suicidal thoughts. Participants responded to the questions using a 4-point Likert scale, with higher scores indicating greater symptom severity. In a previous study, the translated Persian version of the BDI had a Cronbach’s alpha of 0.82 [67].

#### 2.5.5. Symptoms of Anxiety

To assess symptoms of anxiety, participants completed the Farsi version [68] of the Beck Anxiety Inventory (BAI) [69]. The BAI consists of 21 items addressing cognitive, emotional, and physical symptoms of anxiety such as fear of unpleasant events, inability to calm down, nervousness, numbness and numbness, and sweating. Answers were given on a 4-point Likert scale ranging from 0 (not at all) to 3 (severely/it bothered me a lot), with higher sum scores reflecting greater symptoms of anxiety. The validity (Cronbach’s alpha = 0.83) measure of the Persian version of the BAI was confirmed [68].

#### 2.5.6. Sleep Quality

To assess sleep quality, participants completed the Farsi version [70,71] of the Pittsburgh Sleep Quality Index (PSQI) [72]. The PSQI consists of seven elements, such as subjective sleep quality, sleep latency, sleep duration, sleep efficiency, sleep disturbances, sleeping pills, and poor daytime functioning. Each question is graded from 0 to 3, and the scores of these seven components are combined to produce a total score ranging from 0 to 21. A higher score indicates a poorer quality of sleep. A score ≥ 5 indicates poor sleep quality. The validity (Cronbach’s alpha = 0.83) measure of the Persian version of the PSQI was confirmed [73].

### 2.6. Statistical Analysis

#### Statistics Were Performed per Protocol

To compare age, height, and weight between participants in the Aphrodite and placebo conditions, three independent *t*-tests were performed, while three Chi-square (X^2^)-tests were performed to compare civil status, the highest educational degree, and current occupation between the two conditions. To check the homogeneity of the demographic characteristics between the two groups, Fisher’s exact test was used. Next, in order to check the normality of the distribution of scores of depression and sexual performance, sleep quality, and anxiety, a one-sample Shapiro–Wilk test was performed. Shapiro–Wilk tests indicated the data for each outcome variable were normal (all *p* > 0.5). Next, four multivariate analyses of variance with repeated measures (mixed ANOVA) were performed with the independent factors Time (baseline—Week 4—Week 8/end of the study), Group (Aphrodite; placebo) and Time by Group-interaction, and sexual dysfunction, depression, anxiety, and sleep disturbances as dependent variables. Post-hoc tests were performed with single *t*-tests and the Bonferroni–Holm corrections for *p*-values. Effect sizes for F-tests (partial eta-squared; ηp^2^) were reported as follows: ηp^2^ < 0.019 = trivial effect size (T); 0.02 < ηp^2^ < 0.059 = small effect size (S); 0.06 < ηp^2^ < 0.139 = medium effect size (M); ηp^2^ > 0.14 = large effect size (L) [73,74]. Statistical analyses were performed with SPSS^®^ 28.0 (IBM Corporation, Armonk, NY, USA) for Apple Mac^®^.

## 3. Results

### 3.1. Sample Characteristics

Table 1 reports the descriptive and inferential statistical indices of sociodemographic information (age, weight, height; highest educational level; current occupation) between participants in the Aphrodite and the placebo conditions. Overall, no descriptively or statistically significant differences were observed (all *p* > 0.5).

### 3.2. Sexual Function, Symptoms of Depression and Anxiety, and Sleep Quality in the Aphrodite and Placebo Conditions

Table 2 reports the descriptive and inferential statistical indices for sexual function, symptoms of depression and anxiety, and sleep quality in the Aphrodite and placebo conditions. All statistical indices are reported in Table 2 and thus not repeated in the text once again.

### 3.3. Sexual Function

#### 3.3.1. Main Effects

Sexual functions increased over time (significant *p*-value and large effect size), but more so in the Aphrodite condition compared to the placebo condition (significant *p*-value and large effect size). Compared to the placebo condition, the Aphrodite condition had higher sexual function scores (significant *p*-value and large effect size).

#### 3.3.2. Post-Hoc Comparisons

Post-hoc calculations showed that within the Aphrodite condition, sexual functions scores increased from baseline to Week 4 (t (21) = −14, *p* = 0.001), and from baseline to Week 8 (end of the study) (t (21) = −19.92, *p* = 0.001). Within the placebo condition, sexual function scores decreased from baseline to Week 4 (t (18) = 2.61, *p* = 0.01) and from baseline to Week 8 (end of the study) (t (18) = 4.14, *p* = 0.001). As a result, the sexual function of the control group not only did not improve, but also deteriorated.

Post-hoc calculations comparing the groups showed that sexual function scores were similar in the Aphrodite and the placebo groups at baseline (t (39) = 0.41, *p* = 0.68). By contrast, at Week 4 (t (39) = 16.02, *p* = 0.001) and at Week 8 (t (39) = 35.22, *p* = 0.001) (see Figure 2), the participants of the Aphrodite group reported higher sexual function scores than the counterparts from the placebo group (Figure 2).

### 3.4. Symptoms of Depression

#### 3.4.1. Main Effects

Depressive symptoms decreased over time (significant *p*-value and large effect size), but more so in the Aphrodite condition compared to the placebo condition (significant *p*-value and large effect size). Compared to the placebo condition, the Aphrodite condition had lower depressive symptom scores (significant *p*-value and large effect size).

#### 3.4.2. Post-Hoc Comparisons

Post-hoc calculations showed that within the Aphrodite group, depression scores decreased from baseline to Week 4 (t (21) = 7.64, *p* = 0.001) and from baseline to Week 8 (t = 7.42, *p* = 0.001). However, within the placebo group, depressive symptom scores did not decrease significantly from baseline to Week 4 (t (18) = 0.89, *p* = 0.38) and from baseline to Week 8 (t (18) = 1.85, *p* = 0.08).

Post-hoc calculations comparing the groups showed that depression scores were similar in the Aphrodite and the placebo groups at baseline (t (39) = 0.69, *p* = 0.49). By contrast, at Week 4 (t (39) = −5.78, *p* = 0.001) and Week 8 (t (39) = −5.92, *p* = 0.001), the participants of the placebo group reported higher depressive symptoms than the counterparts from the Aphrodite group (Figure 3).

### 3.5. Symptoms of Anxiety

#### 3.5.1. Main Effects

Symptoms of anxiety decreased over time (significant *p*-value and large effect size), but more so in the Aphrodite condition compared to the placebo condition (significant *p*-value and large effect size). Compared to the placebo condition, the Aphrodite condition had lower symptoms of anxiety scores (significant *p*-value and large effect size).

#### 3.5.2. Post-Hoc Comparisons

Post-hoc calculations showed that within the Aphrodite group, anxiety scores decreased from baseline to Week 4 (t (21) = 9.28, *p* = 0.001) and from baseline to Week 8 (t (21) = 8.50, *p* = 0.001). Within the placebo group, anxiety scores decreased substantially from baseline to Week 4 (t (18) = 4.10, *p* = 0.001) and from baseline to Week 8 (t (18) = 4.72, *p* = 0.001).

Post-hoc calculations comparing the groups showed that anxiety scores were similar in the Aphrodite and the placebo groups at baseline (t (39) = −0.44, *p* = 0.65). By contrast, at Week 4 (t (39) = −8.08, *p* = 0.001) and Week 8 (t (39) = −4.03, *p* = 0.001), the participants of the placebo group reported higher anxiety scores than the counterparts from the Aphrodite group (Figure 4).

### 3.6. Sleep Quality

#### 3.6.1. Main Effects

Sleep problems decreased over time (significant *p*-value and large effect size), but more so in the Aphrodite condition compared to the placebo condition (significant *p*-value and large effect size). Compared to the placebo condition, the Aphrodite condition had lower sleep problem scores (significant *p*-value and large effect size).

#### 3.6.2. Post-Hoc Comparisons

Post-hoc calculations showed that within the Aphrodite group, PSQI scores decreased from baseline to Week 4 (t (21) = 4.15, *p* = 0.001), but not from baseline to Week 8 (t (21) = 1.65, *p* = 0.11). Within the placebo group, PSQI scores decreased substantially from baseline to Week 4 (t (18) = −2.70, *p* = 0.01) and from baseline to Week 8 (t (18) = −2.38, *p* = 0.02).

Post-hoc calculations comparing the groups showed that PSQI scores were similar in the Aphrodite and the placebo groups at baseline (t (39) = 0.30, *p* = 0.76). By contrast, at Week 4 (t (39) = −5.60, *p* = 0.001) and Week 8 (t (39) = −3.76, *p* = 0.001), the participants of the placebo group reported higher PSQI scores than the counterparts from the Aphrodite group (Figure 5).

## 4. Discussion

The aim of the present randomized, double-blind, and placebo-controlled study was to compare the influence of Aphrodite (a blend of ginger, saffron, cinnamon, thistle, and *Tribulus terrestris*) on SSRI-induced sexual dysfunctions and symptoms of depression, anxiety, and sleep disturbances in females with major depressive disorder. The overall result was that compared to placebo, Aphrodite improved sexual function and symptoms of depression, anxiety, and—to some extent—sleep disturbances over a time-lapse of eight weeks. Further, and importantly, and compared to placebo, Aphrodite appeared to have the power to counterbalance sertraline-induced sexual dysfunctions. Accordingly, the present findings add to the current literature in an important way that Aphrodite appears to be successfully applicable to females with MDD. Our findings are relevant because females with depression and treated with SSRIs are at increased risk for sexual problems over time.

Here, we discuss the results separately for symptoms. Further, given that we did not assess the dimensions of psychoneuroendocrine changes, we discuss the results based on previous study results.

### 4.1. Sexual Dysfunctions

As regards sexual dysfunctions, our findings are consistent with previous studies. To illustrate, Taavoni et al. [46] showed the effectiveness of Aphrodite on orgasm and sexual desire among postmenopausal women and always compared to placebo. Similarly, Kashani et al. [75] showed that compared to placebo, saffron (30 mg per day) could enhance sexual arousal, lubrication, and the overall sexual function scores among women with major depressive disorder.

To explain the improvement of sexual function following the consumption of Aphrodite, it appeared that the increase of testosterone was key: Gama et al. [76] found that *Tribulus terrestris* could improve serum testosterone and free testosterone levels and consequently total sexual function scores. Further, the results of our study were consistent with Vale et al. [77], who showed thistle as a part of Aphrodite played a decisive role in sexual desire and function among postmenopausal women with symptoms of decreased libido. Sexual function questionnaires like our research were used to evaluate sexual dysfunction before and after treatment. Premenopausal women with hypoactive sexual desire disorder treated with milk thistle experienced an improvement in the total score of sexual function and the domains of “desire”, “sexual arousal”, “lubricate”, “orgasm”, “pain”, and “satisfaction”. In addition, the levels of free and bioavailable testosterone were increased in women who received the herbal pills. In conclusion, it can be argued that herbal medicine may be a safe alternative for the treatment of women with sexual dysfunction, as it is effective in reducing symptoms, possibly due to an increase in the serum level of free and bioavailable testosterone.

In general, the physiological mechanism underlying the efficacy of the herbal remedy on sexual function in women is believed to be related to its impact on hormonal regulation and neurotransmitter activity such as serotonin, dopamine, and norepinephrine [78,79]. These neurotransmitters play a crucial role in regulating mood, emotions, and sexual desire [79]. Additionally, the herbal remedy is known to have adaptogenic properties, helping to balance hormone levels and reduce stress, which can further contribute to improved sexual function [80]. Moreover, the herbal remedy’s antioxidant effects may also protect vascular health and enhance blood flow to genital areas, promoting overall sexual satisfaction and performance in women [81].

Overall, the present results expand upon the current literature in that we showed that Aphrodite improved sexual function among females with MDD and SSRI-induced sexual dysfunction and compared to placebo.

### 4.2. Symptoms of Depression

Our results showed that the Aphrodite decreased symptoms of depression as well. The physiological mechanism underlying the efficacy of herbal remedies in improving depression is believed to involve the modulation of neurotransmitter levels in the brain. Saffron contains bioactive compounds such as crocin and safranal, which are thought to enhance serotonin and dopamine levels, two neurotransmitters that play a critical role in regulating mood and emotions. By increasing the availability of these neurotransmitters, saffron may help alleviate symptoms of depression [82].

Previous studies showed that neurotropic agents play an important role in the pathogenesis of depression as well as depression treatments [83]. Among the most widespread neurotrophic factors is brain-derived neurotrophic factor (BDNF), which is expressed in the hippocampus as well as the cerebral cortex [84,85]. Clinical and pharmacological studies showed that BDNF concentration decreases in the serum of patients with MDD. A previous study showed that the ginger herbal pill has an antidepressant effect [86], which is part of Aphrodite. In this line, Xue et al. [87] showed that the ginger plant rapidly increased hippocampal BDNF expression. Lechtenberg et al. [88] also showed that saffron showed good affinity for the phencyclidine (PCP) binding site of the NMDA receptor to build BDNF.

### 4.3. Symptoms of Anxiety

Our results showed that Aphrodite decreased the symptoms of anxiety compared to placebo. Aphrodite herbal contains bioactive compounds such as gingerol and zingerone, which have been shown to exert anxiolytic effects. These compounds are thought to interact with gamma-aminobutyric acid (GABA) receptors in the brain, leading to increased GABAergic activity. GABA is an inhibitory neurotransmitter that helps regulate anxiety and stress responses by reducing neuronal excitability [89]. By enhancing GABAergic neurotransmission, ginger may promote relaxation and decrease anxiety levels. On the other hand, saffron contains crocin and safranal, which have been shown to influence serotonin levels in the brain. Serotonin is a neurotransmitter associated with mood regulation, and its dysregulation is often linked to anxiety disorders. Saffron’s bioactive compounds are believed to enhance serotonin availability by inhibiting its reuptake, thereby prolonging its presence in synaptic spaces [90]. This increased serotonin transmission may contribute to the anxiolytic effects of saffron, promoting a sense of calmness and reducing anxiety symptoms.

Moreover, both ginger and saffron possess antioxidant and anti-inflammatory properties, which can protect the brain from oxidative stress and neuroinflammation [91]. Chronic inflammation and oxidative damage have been implicated in anxiety disorders, and the ability of these herbal remedies to combat these processes may further contribute to their anxiolytic effects [92].

Overall, the present results expand upon the current literature in that we showed that Aphrodite improved symptoms of depression and anxiety among females with MDD and SSRI-induced sexual dysfunction and compared to placebo.

### 4.4. Sleep Disturbances

Another part of our results showed that Aphrodite reduced sleep problems. Current evidence suggests that the neurobiological mechanism of sleep problems is associated with dysregulation of serotonergic, noradrenergic, glutamatergic, and GABAergic transmission. Saffron has also been mentioned as one of the plants effective in improving sleep. Previous studies have shown that saffron has effects on relaxation of the central nervous system [93].

### 4.5. Psychiatric and Psychological Considerations

Besides the neuroendocrinological models proposed above, two psychological concepts are considered to explain as to why apparently symptoms of sexual dysfunction, depression, anxiety, and sleep disturbances changed concomitantly and in parallel. The first is the concept of the transdiagnostic approach; the second is the concept of allostatic load.

First, the transdiagnostic approach in psychiatry and clinical psychology [92,94,95,96,97] observed that the clinical presentation of individuals with symptoms of depression is such that they also report symptoms of anxiety and insomnia. In the same vein, and based on clinical and epidemiological surveys, approximately half of patients with the principal diagnosis of anxiety disorder also meet the criteria for at least one additional comorbid psychiatric disorder. Given this, it appears plausible that symptoms of sexual dysfunction, depression, anxiety, and insomnia were reported concomitantly.

Second, the concept of allostatic load in well-being and ill-being suggests that a person per sé might not suffer (well-being) or suffer (ill-being) from psychological distress [98]. In a more specific context, allostatic overload is understood as the cumulative effects of stressful experiences in daily life and may lead to disease over time [98,99,100,101]. Given this background, it appears plausible that improvements in sexual function decreased the allostatic load, including decreased symptoms of depression, anxiety, and sleep disturbances.

### 4.6. Strengths

The strengths of the study are (1) the randomized, double-blind, and placebo-controlled study design, including a thorough and reliable psychiatric assessment and clearly defined inclusion and exclusion criteria; (2) the inclusion of standardized and internationally validated self-rating questionnaires; (3) the assessment of several important psychological factors such as depression, anxiety, and subjective sleep; and (4) the three time-points.

### 4.7. Limitations

The novel and intriguing results of the present study should be balanced against the following limitations. First, the research was performed in one study center in Kermanshah province, and thus a systematic center-related bias cannot be excluded. Second, participants’ diet was not assessed, while in the meanwhile there is sufficient evidence that behavioral health disorders like depression and anxiety appeared to be associated with nutritional compounds such as probiotics [102]. Due to the critical function that nutrients may play on the neuroendocrine system, poor nutrition has been linked to the pathology at the cause of behavioral health disorders. As a result, a poor diet that provides insufficient nutrient intake is considered a risk factor for depression [101]. Third, while results showed that Aphrodite had a higher favorable influence, when compared to placebo, the quality of the data did not allow any insights as regards the possible underlying neurophysiological and neuroendocrine changes. More specifically, the assessment of salivary cortisol, blood BDNF concentrations and objectively assessed sleep parameters would have helped to understand the psychophysiological basis of the present results. In this regard, we note that higher cortisol concentrations, lower BDNF concentrations, and disrupted sleep are considered reliable biomarkers of psychophysiological arousal and of symptoms of depression [103]. Fourth, following this observation, it is conceivable that other latent but unassessed psychological factors such as higher Aphrodite-induced cognitive functions or an improved self-efficacy [104] might have biased two or more dimensions in the same or opposite directions. Further possible confounders might have been the duration of marriage, the number of children and pregnancies, and elective or urgent cesarean sections compared to vaginal deliveries, along with the overall satisfaction of the marital and sexual life. All these confounders might have biased the sample from the very beginning during the randomization process. Fifth, one might claim that the sample size was rather small, though, we calculated the sample size a priori, we prevalently relied on effect size calculations, which are virtually not sensitive to sample sizes, and we considered that relying on mere *p*-values is dated [105]. Last, we assessed exclusively female participants due to the constraints of the research design and due to cultural considerations in Iran.

Given these limitations, future studies should assess both males and females, assess further psychological and neurophysiological variables, and ask about participants’ overall sexual and marital satisfaction, including marital indices such as the duration and quality of couple life. Such a thorough assessment might also include the number of children, deliveries, miscarriages, and the quality of pregnancy and delivery [106]. Next, a further plus will be a follow-up assessment some weeks and months after the study’s end. Ideally, such a follow-up considers if and to what extent the partner (here the husband) would rate the overall quality of their sexual and marital life.

## 5. Conclusions

Aphrodite has been found to improve sexual function and sleep quality and reduce depression and anxiety symptoms. It has the potential to be prescribed as a medicinal plant for the treatment of patients with MDD and related symptoms. However, further research is required to fully understand its mechanism of action and potential side effects. More experimental and longitudinal studies are needed to examine the effect of Aphrodite on other variables related to sexual function and the quality of sexual relations between couples. Additionally, investigating other variables such as testosterone levels may provide more insight into the nature and the effict of Aphrodite.

## Figures and Tables

**Figure 1 medicina-59-01663-f001:**
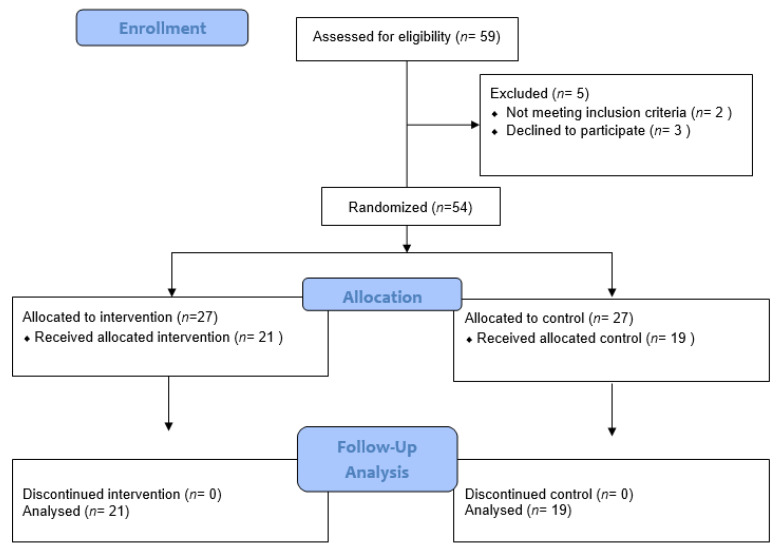
Flow chart of the number of participants, including their condition assignments.

**Figure 2 medicina-59-01663-f002:**
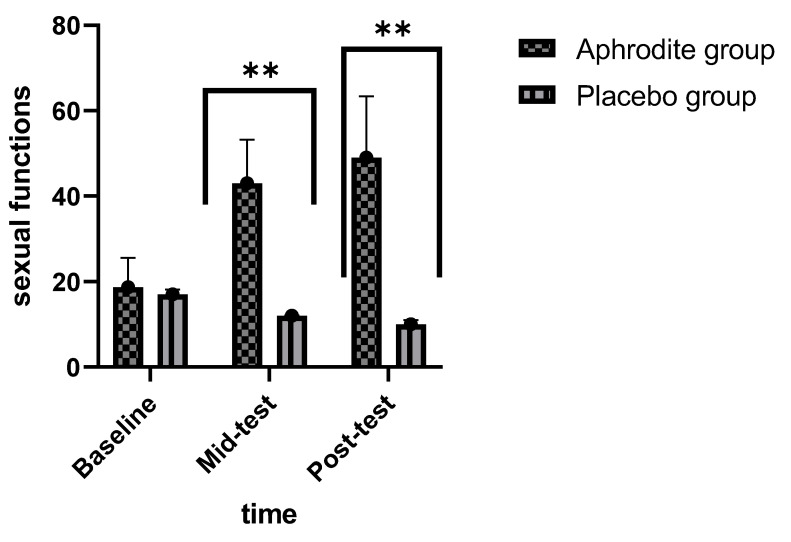
Comparisons of the Aphrodite and placebo groups from baseline, Week 4, and Week 8 for sexual functions (comparing the groups; ** *p* < 0.01). Bars are means; lines are standard deviations.

**Figure 3 medicina-59-01663-f003:**
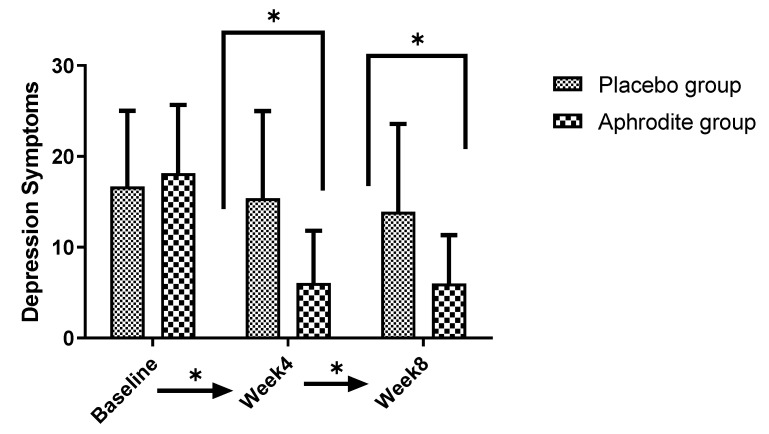
Comparisons of the Aphrodite and placebo groups from baseline, Week 4, and Week 8 for depression symptoms (comparing the groups; * *p* < 0.05). Bars are means; lines are standard deviations.

**Figure 4 medicina-59-01663-f004:**
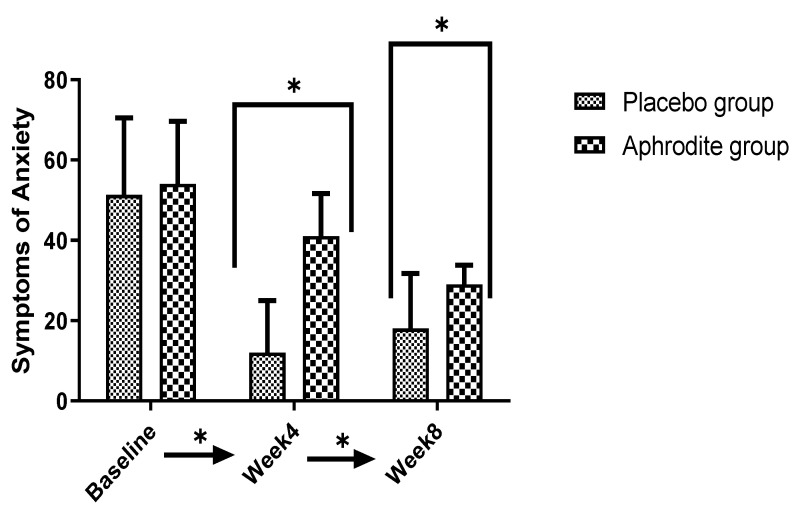
Comparisons of the Aphrodite and placebo groups from baseline, Week 4, and Week 8 for anxiety symptoms of anxiety (comparing the groups; * *p* < 0.05). Bars are means; lines are standard deviations.

**Figure 5 medicina-59-01663-f005:**
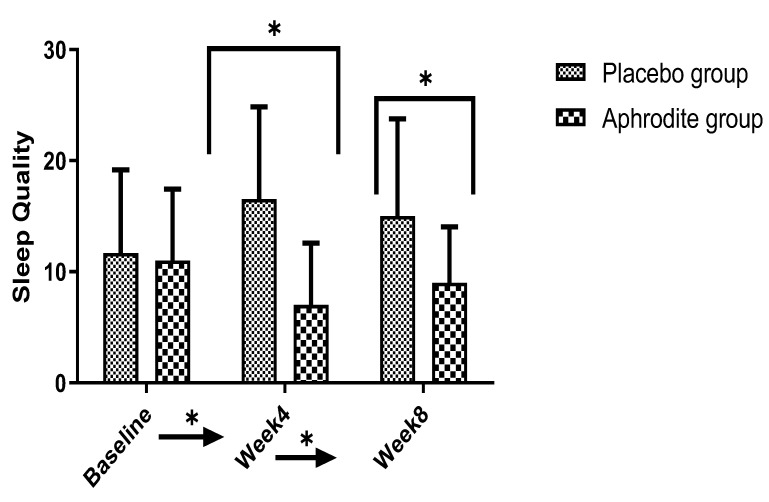
Comparisons of the Aphrodite and placebo groups from baseline, Week 4, and Week 8 for sleep quality (comparing the groups; * *p* < 0.05). Bars are means; lines are standard deviations.

**Table 1 medicina-59-01663-t001:** Descriptive statistics and overview of sociodemographic background, separately for Aphrodite and placebo at baseline.

	Group		Statistics
Dimensions	Placebo	Aphrodite	
N	19	21	
	Mean (SD)	Mean (SD)	
Age (years)	34.47 (8.79)	36.64 (8.35)	t (39) = 1.89, *p* = 0.08
Weight (kg)	66.89 (10.39)	69.36 (13.48)	t (39) = 0.97, *p* = 0.14
Height (cm)	160.63 (4.91)	162.63 (6.26)	t (39) = 0.82, *p* = 0.16
	*n* (%)	*n* (%)	
Occupational status			X2(N = 41; df = 1) =0.02, *p* = 0.91
Unemployed	13	11	
Employed	6	11	
Educational level			X2(N = 41; df = 3) =0.5, *p* = 0.77
Middle school	5	8	
High school	5	3	
Bachelor	8	10	
Master	1	0	

Notes: N: number, M: mean, SD: Standard deviation, cm: centimeters, Kg: kilogram.

**Table 2 medicina-59-01663-t002:** Descriptive and inferential statistical indices for sexual function, depressive symptoms, sleep quality, and anxiety at baseline at Week 4 and Week 8 and separately for the Aphrodite and control conditions.

	Aphrodite (*n* = 21)		Placebo (*n* = 19)		Factors					
					Group		Time		Time × Group Interaction	
Variable	*M*	*SD*	*M*	*SD*	* F *	η_p_^2^	* F *	η_p_^2^	* F *	η_p_^2^
Sexual function					121.07 *	0.71	60.18 **	0.60	150.90 **	0.79
Baseline	18.86	6.60	17.94	7.69						
Week 4	43.95	7.37	12.94	4.39						
Week 8/study end	49.63	2.42	10.63	4.49						
Depressive symptoms					16.13 **	0.35	29.62 **	0.49	13.90 **	0.34
Baseline	18.13	6.85	16.57	7.48						
Week 4	6.54	3.93	15.52	6.56						
Week 8/study end	6.18	3.68	13.21	3.90						
Anxiety					20.29 **	0.35	67.77 **	0.63	11.91 **	0.23
Baseline	51.68	19.94	54.52	20.62						
Week 4	12.54	9.44	41.68	13.52						
Week 8/study end	18.27	9.36	29.42	8.16						
Sleep quality					19.75 **	0.33	0.85	0.004	11.60 **	0.22
Baseline	11.81	5.51	11.31	4.89						
Week 4	7.09	4.25	16.47	6.38						
Week 8/study end	9.00	5.35	15.31	5.36						

Notes: M: mean, SD: Standard deviation, * *p* < 0.05.; ** *p* < 0.01.

## Data Availability

Data is made available to explicit experts in the field. Such experts should clearly formulate their hypotheses; further, they should fully describe, how and where they do securely store the data file, and how they make sure that the data file is not shared with and securely protected from third parties.

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
