# Peer review of "Effects of Aphrodite (an Herbal Compound) on SSRI-Induced Sexual Dysfunctions and Depression in Females with Major Depressive Disorder: Findings from a Randomized Clinical Trial"

_medicina, 2023, doi:10.3390/medicina59091663_

Round 1

Reviewer 1 Report

This RCT assessed the impact of Aphrodite on SSRI-induced sexual dysfunctions and depression in females with major depressive disorder.

Major

The randomization process did not consider several marital factors, such as the duration of the marriage, maternal factors, such as the number of children/pregnancies and history of C/S or vaginal tear, and other factors, such as FGM. This could be partly understood given the small sample size; however, adjusting for these variables during the analysis should have been considered.

Besides, it seems that the results did not adjust for age and BMI.

English editing is needed.

More discussions are needed in the limitation section (please add title strengths and limitations)

The footnotes of the tables and figures should show the statistical analyses used and adjusted variables.

Table 1 should show which variables were presented as n (%) and which variables were presented as mean (SD). P-values to show the differences across groups are needed.

 Minor

In lines 475 and 480, as well as other parts, the references were not written per the journal requirements.

English editing is recommended.

Author Response

We thank Reviewer #1 for their valuable comments, which helped us to improve the quality of the revision. Please find the detailed point-by-point-response attached as a separate file. 

Thank you again for all your kind efforts.

Reviewer 2 Report

The manuscript by Shahmoradi et al. describes the effects of Aphrodite herbal drug in women diagnosed with MDD and sexual dysfunction induced by SSRIs. The results showed that, compared with a placebo, the treatment with Aphrodite improved the sexual functioning and sleep quality of females also reducing depression and anxiety scores. There are some issues that need to be addressed:

1. Title
• The title is too long and needs improvement to achieve clarity.
• It might be considered not to specify the content of Aphrodite and instead
refer to it as an "herbal drug" or “herbal compound” (effects of the herbal
drug Aphrodite…).
• Phrasing is confusing and suggests that SSRIs induce the depressive
symptoms to be addressed.
• Instead of the semicolon there should be a colon before “results from a
randomized…”.
• Check this: Tribulus terrestris (gender must be written in initial capital letter
and the entire name in italics across the document).

2. Abstract
Several words are repeated, making it redundant. There is very specific information regarding the results that could be omitted.

3. Introduction
Improve the introduction, especially the language.
• Improve in-text citations (eg: check this: internet-delivered, on line 61)
• Check this: various literatures, tribulus terrestris, given (lines 132-134)
• Lines 113-114: It is not clear whether the results correspond to the isolated or
combined effect of the components.
• No need to specify components every time Aphrodite is mentioned.
• Lines 124-128: There is no need to specify the statistical values of the
referenced study.
• The hypothesis is not well written, and it should not appear as a separate line.
• In the introduction nothing is mentioned about the quality of sleep and its
role in depression and sexual functioning.

4. Methods
• It is not justified why the study was carried out only on women, even when
in the introduction the authors mentioned that “47.7% of men and 15.4% of
women had discontinued treatment due to sexual effects from psychiatric
medications”.
• In the inclusion criteria it is not clear if the participants are under a
therapeutic regimen just with sertraline or combined drugs.
• Were all participants taking the same therapeutic dosages of sertraline? It
seems like 50 to 200 mg is a wide range and could add heterogeneity to the
sample.
• The flow chart of participants needs revision. Eg: Other reasons (n=0) should
be omitted; excluded from the analysis (give reasons) = 0 should be omitted.
It is not clear what the authors mean by “Did not receive allocated
intervention/control”. Follow-up and analysis could be merged at the same
level.
• The section called “medications” is confusing. It seems like sertraline was
another treatment (not an inclusion criterion).
• How many Aphrodite tablets the participants consumed daily? In the
abstract, it is mentioned that it was two, but in this section, it is not clarified.
• Check this: “in shape, weight, color, and shape”.
• What do the authors mean by a thorough psychiatric interview? They must
provide details regarding the specific clinical standardized interview that was
used to perform the diagnosis.
• Regarding SSRIs-indued sexual dysfunction it is not clear whether the
authors performed a longitudinal assessment or if it was determined
retrospectively by self-reports.
• There are some outstanding issues regarding the statistical analysis that need
to be revised:
o Specified the meaning of the X2-test
o Authors mentioned that X2-test was used to compare gender between
conditions, but all participants were women.
o The authors mentioned that X2-tests and T-tests were used to analyze
demographic factors. However, they also mentioned that F-test and X2
test were used to check the homogeneity of those factors. It is
confusing.
o What do the authors mean by General Linear Model-Repeated Model?
generalized linear model repeated measures?
o It is not clear if week 8 is the same as “end of the study” or corresponds
to another time interval.

5. Results
• Check this: all ps (line 277)
• Authors must provide captions to the tables.
• M should be replaced by mean
• In table 1 the authors did not include the statistical values for many factors
(eg: weight, height…)
• The authors did not include the results of Shapiro-Wilk test
• In the methods section the authors mentioned that they used a generalized
linear model for analyzing sexual function, depressive symptoms, and sleep
quality. However, in the results, they showed statistical values corresponding
to F-test. It is not clear. Also, there is no captions in the tables. Besides, authors
should include p-values.
• In Figure 2 the group conventions are the same. It does not include differences
between week 4-8 and baseline.
• The units of measurement of the dependent variables are missing in all
figures.

6. Discussion
• Improve the language.
• It would be convenient not to break it down into sections.
• Standardize the citation format within the text and in references.
• Citations are missing in some paragraphs.
• Check this: neurotropic (line 426).
• What is the meaning of PCP? (line 434).

7. References
• Replace old references with recent ones, e.g., Maag et al., 2005; Gibbons et al.,
2011; Rosenberg et al., 2003; Kessler et al., 2002; Shokrollahi et al., 1999; Duman
et al., 1997; Smith et al., 1995.
• Homogenize format.

-

Author Response

We thank Reviewer #2 for their valuable comments, which helped us to improve the quality of the revision. Please find the detailed point-by-point-response attached as a separate file. 

Thank you again for all your kind efforts.

Round 2

Reviewer 1 Report

The manuscript has sufficiently improved.

Minor corrections and modifications can be made during the final English editing step.